# Genetic Association of *APOA5* and *AKT3* Genes with Milk Production Traits in Chinese Holstein Cows

**Zijiao Guo [1], Aixia Du [1], Bo Han [1], Hui Li [2], Rugang Tian [2], Wei Sun [3], Gaoping Zhao [3], Jing Tian [2], Xiangnan Bao [3], Jixin Zhang [4], Lingna Xu [1] and Dongxiao Sun [1,\***

[1] Department of Animal Genetics and Breeding, College of Animal Science and Technology, Key Laboratory of Animal Genetics, National Engineering Laboratory for Animal Breeding, State Key Laboratory of Animal Biotech Breeding, China Agricultural University, Breeding and Reproduction of Ministry of Agriculture and Rural Affairs, Beijing 100193, China; gzjiao@cau.edu.cn (Z.G.); duaixia@cau.edu.cn (A.D.); bohan@cau.edu.cn (B.H.); xulingna@caas.cn (L.X.)

[2] Inner Mongolia Academy of Agricultural and Animal Husbandry Sciences, Hohhot 010031, China; lihuizh@126.com (H.L.); tiannky@163.com (R.T.); tianj729@163.com (J.T.)

[3] Inner Mongolia SK Xing Animal Breeding and Breeding Biotechnology Research Institute Co., Ltd., Hohhot 011517, China; swzyh769500@163.com (W.S.); gaopingzhao@126.com (G.Z.); 18548163596@163.com (X.B.)

[4] Inner Mongolia XuYi Animal Husbandry Co., Ltd., Bayannur 015000, China; 18047816669@163.com

\* Correspondence: sundx@cau.edu.cn

**Abstract:** Genome selection (GS) technology is an important means to improve the genetic improvement of dairy cows, and the mining and application of functional genes and loci for important traits is one of the important bases for accelerating genetic improvement. Our previous study found that the apolipoprotein A5 (*APOA5*) and AKT serine/threonine kinase 3 (*AKT3*) genes were differentially expressed in the liver tissue of Chinese Holstein cows at different lactation stages and influenced milk component synthesis and metabolism, so we considered these two genes as the candidates affecting milk production traits. In this study, we found in total six single nucleotide polymorphisms (SNPs), three in *APOA5* and three in *AKT3*. Subsequent association analysis showed that the six SNPs were significantly associated with milk yield, fat yield, protein yield, or fat percentage ($p \leq$ 0.05). Three SNPs in *APOA5* formed a haplotype block, which was found to be significantly associated with milk yield, fat yield, and protein yield ($p \leq 0.05$). In addition, four SNPs were proposed to be functional mutations affecting the milk production phenotype, of which three, 15:g.27446527C>T and 15:g.27447741A>G in *APOA5* and 16:g.33367767T>C in *AKT3*, might change the transcription factor binding sites (TFBSs), and one is a missense mutation, 15:g.27445825T>C in *APOA5*, which could alter the secondary structure and stability of mRNA and protein. In summary, we demonstrated the genetic effects of *APOA5* and *AKT3* on milk production traits, and the valuable SNPs could be used as available genetic markers for dairy cattle's GS.

**Keywords:** *APOA5*; *AKT3*; milk production traits; association analysis; SNP

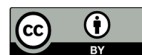

## 1. Introduction

Milk is a naturally nutritious food that can provide a plethora of essential nutrients including high-quality proteins, fats, carbohydrates (lactose), minerals, trace elements, and vitamins for the human diet [1,2]. Beyond its nutritional value, mounting evidence suggests that milk may confer numerous health-related benefits; these include its potential roles in preventing cardiovascular diseases, cancers, obesity, and diabetes, among others [3–6]. With the improvement of economic levels and awareness of nutrition and health, milk consumption will undoubtedly continue to increase, so it is crucial to enhance milk production and its nutritional content.

In recent years, researchers have implemented multiple strategies to improve the production performance of dairy cattle. Among them, genomic selection (GS) using dense

markers covering the whole genome is a strategy for the genetic improvement of livestock and has revolutionized the breeding system in dairy cattle [7–9]. Since 2009, GS technology has been widely used in early selective breeding of dairy cattle, which significantly shortens generation intervals and reduces breeding costs, accelerating genetic progress within the population [10,11]. Studies have shown that incorporating single nucleotide polymorphism (SNP) information from functional genes with significant genetic effects on the target traits into chip marker data can enhance the accuracy of genomic estimated breeding values [12,13]. Therefore, more and more studies are dedicated to identifying functional genes and SNP loci that have a significant impact on milk production traits [14–19], with the aim of applying them to dairy cattle's GS to improve the accuracy of milk production trait selection.

Previously, we analyzed the proteomes of liver tissue samples from three Holstein cows during the dry period and early and peak lactations and found that the apolipoprotein A5 (*APOA5*) and AKT serine/threonine kinase 3 (*AKT3*) genes exhibited differential expression across various lactation stages, and they were also involved in the pathways related to the synthesis and metabolism of milk components, so these two genes were considered to be promising candidate genes that affect milk production traits [20]. The *APOA5* gene is an integral part of the regulation of plasma triglyceride levels [21–23] and is highly expressed in the liver of periparturient cows, regulating the synthesis and metabolism of fatty acids and lipoproteins in preparation for lactation [24,25]. *AKT3* is a major nodal gene in the phosphatidylinositol 3-kinase (PI3K)/Akt pathway, which regulates cell proliferation, differentiation, apoptosis, and other biological processes by responding to extracellular signals [26,27]. It also participates in mammalian target of rapamycin (mTOR), AMP-activated protein kinase (AMPK), and insulin receptor signaling networks, which are the pathways related to the lactation of dairy cows [28]. In addition, the *APOA5* gene was found to be located 0.22–2.13 Mb away from the quantitative trait loci (QTL) associated with fat yield and percentage, protein yield and percentage, and fatty acid content [29–32]. *AKT3* was located near 2.03–3.33 Mb of known QTLs for milk yield and fat yield [30,33]. Therefore, we considered that the *APOA5* and *AKT3* genes might play important roles for milk production traits in dairy cattle.

In this study, we identified the single nucleotide polymorphisms (SNPs) of the *APOA5* and *AKT3* genes in a Chinese Holstein population and analyzed their genetic associations with 305-day milk yield, fat yield, fat percentage, protein yield, and protein percentage. Further, we conducted functional predictions of key mutation sites to speculate on the reasons why they affect milk production traits. The purpose of this study is to provide valuable SNP loci information for dairy cattle's GS and gene information for the in-depth study of the mechanism related to milk production traits in dairy cattle.

## 2. Materials and Methods

### 2.1. Animals and Phenotypic Data

In this study, a total of 944 Chinese Holstein cows in the first lactation and 637 in the second lactation (307 cows had just finished the milking of first lactation) were used for association analyses. The cows were from 45 sire families and fed under the same conditions in 22 dairy farms of Beijing Sunlon Livestock Development Co., Ltd. (Beijing, China), where each sire family had 1–68 daughters, with an average of 21. Each cow had pedigree information and dairy herd improvement (DHI) records, which were provided by the Beijing Dairy Cattle Center (Beijing, China). The descriptive statistics of phenotypic values for milk production traits of the first and second lactations are presented in Table S1.

### 2.2. DNA Extraction and Quality Control

DNA was extracted from semen samples of the 45 sires using the salt-out procedure and from blood samples of the 944 cows with a TIANamp Blood DNA Kit (Tiangen, Beijing, China). These frozen semen and blood samples were provided by Beijing Dairy Cattle

Center. Then, a NanoDrop 2000 Spectrophotometer (Thermo Scientific, Hudson, NH, USA) and gel electrophoresis were used to determine the quantity and quality of the extracted DNA, respectively.

### 2.3. SNP Identification and Genotyping

According to the sequences of bovine *APOA5* (Gene ID: 538914) and *AKT3* (Gene ID: 100137872) downloaded from GenBank (https://www.ncbi.nlm.nih.gov/genbank/, accessed on 12 January 2024), primers were designed by Primer3 (https://primer3.ut.ee/, accessed on 12 January 2024) to amplify these genes' coding regions and 2000 bp of upstream and downstream flanking regions (Table S2). The primers were synthesized by BGI Genomics Co., Ltd. (Beijing, China). The DNA samples of the 45 bulls were used as the template for PCR amplification (Table S2), and then its products were sequenced by Sanger sequencing. After that, the potential SNPs were identified by comparing the sequences with the reference sequence (ARS-UCD1.2) through NCBI-BLAST (https://blast.ncbi.nlm.nih.gov/Blast.cgi, accessed on 12 January 2024). Subsequently, the identified SNPs were genotyped in the 944 cows using Genotyping by Target Sequencing (GBTS) technology by Boruidi Biotechnology Co., Ltd. (Shijiazhuang, China).

### 2.4. Association Analyses

Haploview4.2 (Broad Institute of MIT and Harvard, Cambridge, MA, USA) was utilized to estimate the extent of linkage disequilibrium (LD) between the identified SNPs, and pairwise SNP correlations were represented by $R^2$ when $R^2 = 1$ indicated that the SNPs were in complete linkage disequilibrium. Then, SAS 9.4 (SAS Institute Inc., Cary, NC, USA) was used to assess the association between the SNPs/haplotype blocks and milk yield and composition traits on the first and second lactations with the following animal model:

$$y_{ijkl} = \mu + HYS_j + b \times M_k + G_i + a_l + e_{ijkl}$$

where $y_{ijkl}$ is the phenotypic value of each trait for each cow; $\mu$ is the overall mean; $HYS_j$ is the fixed effect of the farm (1–22 for 22 farms), year (1–4 for the years 2012–2015, respectively), and season (1 for April–May; 2 for June–August; 3 for September–November; and 4 for December–March); $M_k$ is the age of calving as a covariant; b is the regression coefficient of covariant M; $G_i$ is the genotype or haplotype combination effect; $a_l$ is the individual random additive genetic effect, distributed as $N(0, A\delta_a^2)$, with the additive genetic variance $\delta_a^2$; and $e_{ijkl}$ is the random residual, distributed as $N(0, I\delta_e^2)$, with identity matrix I and residual error variance $\delta_e^2$. Multiple tests were implemented by Bonferroni correction, with the significance level equal to the original *p* value multiplied by the number of genotype or haplotype combinations.

In addition, the additive effect (a), dominant effect (d), and substitution effect ($\alpha$) were calculated using the following formulas: $a = \frac{AA - BB}{2}$, $d = AB - \frac{AA + BB}{2}$, $\alpha = a + d(q - p)$, where AA, BB, and AB are the least squares means of the milk production traits in the corresponding genotypes, p is the frequency of allele A, and q is the frequency of allele B.

### 2.5. Functional Prediction of Mutation Sites

The Jaspar online website (http://jaspar.genereg.net/, accessed on 20 March 2024) was employed to predict whether SNPs in the 5′ flanking region of the *APOA5* and *AKT3* genes changed the transcription factor binding sites (TFBSs; relative score (RS) ≥ 0.85). To predict changes in mRNA secondary structures for missense mutation, RNAfold web server (http://rna.tbi.univie.ac.at/cgi-bin/RNAWebSuite/RNAfold.cgi, accessed on 20 March 2024) was used, with the minimum free energy (MFE) of the optimal secondary structure reflecting the stability of the mRNA structure. A lower MFE value indicates greater stability in the mRNA structure. Additionally, the impact of missense mutation on protein sec-

ondary structure, including α-helix, β-turn, extended strand, and random coil, was determined using SOPMA (https://npsa-pbil.ibcp.fr/cgi-bin/npsa_automat.pl?page=/NPSA/npsa_sopma.html, accessed on 20 March 2024). Changes in protein stability caused by mutation were predicted through SAAFEC-SEQ Web (http://comp-bio.clemson.edu/SAAFEC-SEQ/, accessed on 20 March 2024). The changes in ΔΔG value before and after mutation represent alterations in protein stability, where a value greater than zero indicates an increase in stability. Finally, PROVEAN (http://provean.jcvi.org/seq_submit.php, accessed on 20 March 2024) was applied to predict whether the protein function was altered before and after the mutation, and when the score was lower than −2.5, it was considered to be a harmful mutation.

## 3. Results

### 3.1. SNP Identification

We found three SNPs in the *APOA5* gene and three in *AKT3*. In *APOA5*, two SNPs, 15:g.27447741A>G (rs41755770) and 15:g.27446527C>T (rs1755767), were located in the 5′ regulatory region and one SNP, 15:g.27445825T>C (rs41755766), in exon 1, a missense mutation in which, when the allele mutates from T to C, the amino acid changes from lysine (AAG) to arginine (AGG). In *AKT3*, 16:g.33367767T>C (rs208316642) was located in the 5′ regulatory region, 16:g.33417238C>T (rs41798799) in intron 1, and 16:g.33551706T>C (rs209739552) in intron 6 (Table 1). The genotypic and allelic frequencies of all the identified SNPs are summarized in Table 1.

**Table 1.** Details of SNPs identified in *APOA5* and *AKT3* genes.

| Gene | SNP Name | GenBank No. | Location | Genotype | Genotypic Frequency | Allele | Allelic Frequency |
|---|---|---|---|---|---|---|---|
| *APOA5* | 15:g.27447741A>G | rs41755770 | 5′ regulatory region | AA | 0.0975 | A | 0.3173 |
| | | | | AG | 0.4396 | G | 0.6827 |
| | | | | GG | 0.4629 | | |
| | 15:g.27446527C>T | rs41755767 | 5′ regulatory region | CC | 0.1070 | C | 0.3332 |
| | | | | CT | 0.4523 | T | 0.6668 |
| | | | | TT | 0.4407 | | |
| | 15:g.27445825T>C (missense mutation) | rs41755766 | exon 1 | CC | 0.4523 | C | 0.6748 |
| | | | | CT | 0.4449 | T | 0.3252 |
| | | | | TT | 0.1028 | | |
| *AKT3* | 16:g.33367767T>C | rs208316642 | 5′ regulatory region | CC | 0.1631 | C | 0.4115 |
| | | | | CT | 0.4968 | T | 0.5885 |
| | | | | TT | 0.3400 | | |
| | 16:g.33417238C>T | rs41798799 | intron 1 | CC | 0.3612 | C | 0.5990 |
| | | | | CT | 0.4756 | T | 0.4010 |
| | | | | TT | 0.1631 | | |
| | 16:g.33551706T>C | rs209739552 | intron 6 | CC | 0.0191 | C | 0.1563 |
| | | | | CT | 0.2744 | T | 0.8438 |
| | | | | TT | 0.7066 | | |

### 3.2. Association Analyses between SNPs and Five Milk Production Traits

We analyzed the genetic associations between the six SNPs in the *APOA5* and *AKT3* genes and five milk production traits, including 305-day milk yield, fat yield, fat percentage, protein yield, and protein percentage (Table S3). In *APOA5*, 15:g.27447741A>G had significant associations with milk yield ($p = 0.0025$) and protein yield ($p = 0.0146$) in the first lactation and milk yield ($p = 0.0003$), fat yield ($p = 0.0019$), and protein yield ($p = 0.0041$) in the second lactation. 15:g.27446527C>T was found to be significantly associated with milk yield ($p = 0.0012$) and protein yield ($p = 0.0108$) in the first lactation and milk yield ($p$

< 0.0001), fat yield (*p* < 0.0001), and protein yield (*p* < 0.0001) in the second lactation. 15:g.27445825T>C was significantly associated with milk yield in the first lactation (*p* = 0.0116) and milk yield (*p* < 0.0001), fat yield (*p* < 0.0001), and protein yield (*p* = 0.0030) in the second lactation.

In *AKT3*, 16:g.33367767T>C was significantly associated with fat yield (*p* = 0.0035) in the first lactation and fat yield (*p* < 0.0001), fat percentage (*p* = 0.0141), and protein yield (*p* = 0.0017) in the second lactation. 16:g.33417238C>T had significant associations with milk yield (*p* = 0.0141), fat yield (*p* = 0.0030), and protein yield (*p* = 0.0003) in the second lactation. 16:g.33551706T>C was found to be significantly associated with milk yield (*p* = 0.0005) and protein yield (*p* < 0.0001) in the second lactation. Additionally, the additive, dominant, and substitution effects of the six SNPs are shown in Table S4.

In *APOA5*, for 15:g.27447741A>G (Figure 1A), 15:g.27446527C>T (Figure 1B,C), and 15:g.27445825T>C (Figure 1D), we observed that the GG, TT, and CC genotypes were the dominant genotypes for milk yield or protein yield. As for the *AKT3* gene, the CC genotype in 16:g.33367767T>C was the dominant genotype for fat yield (Figure 1E).

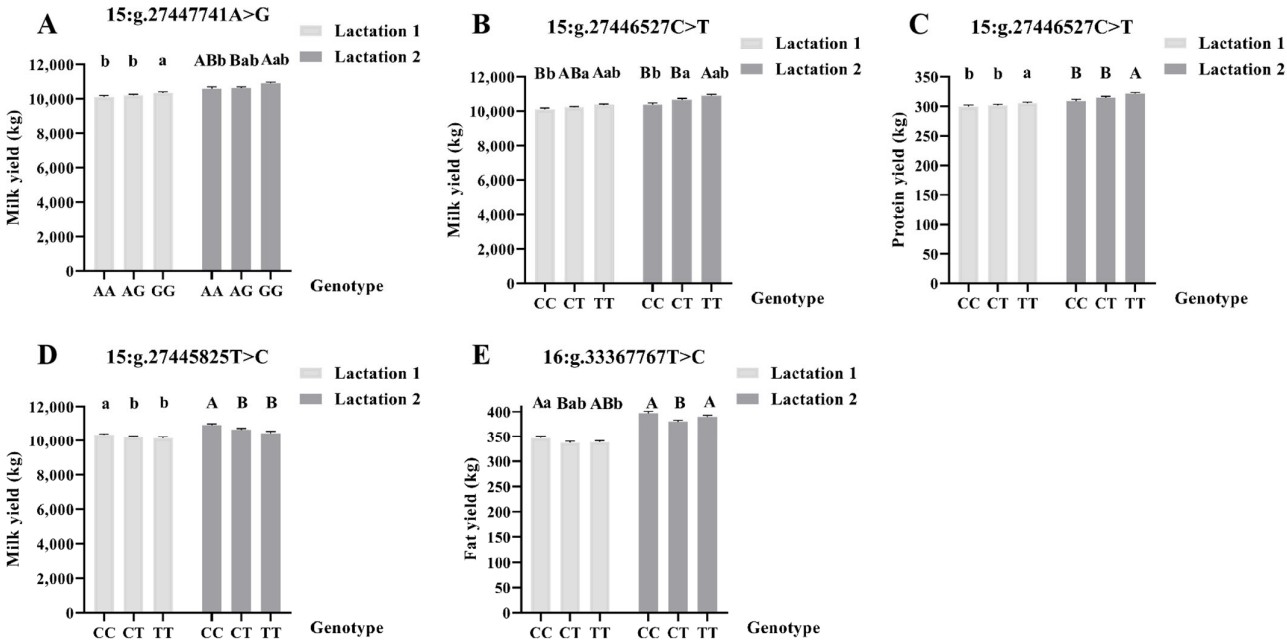

**Figure 1.** Phenotypes of milk production traits for genotypes in different SNPs. (**A**) Milk yield of different genotypes in 15:g.27447741A>G; (**B**) milk yield of different genotypes in 15:g.27446527C>T; (**C**) protein yield of different genotypes in 15:g.27446527C>T; (**D**) milk yield of different genotypes in 15:g.27445825T>C; (**E**) fat yield of different genotypes in 16:g.33367767T>C. a or b indicate significant differences between the phenotypes of milk production traits of different genotypes ( *p* ≤ 0.05); A or B indicate extremely significant differences between the phenotypes of milk production traits of different genotypes ( *p* ≤ 0.01).

### 3.3. Association between Haplotype Block and Five Milk Production Traits

We estimated the degree of LD among the identified SNPs in *APOA5* and *AKT3* using Haploview4.2 and found that one haplotype block including three SNPs, 15:g.27447741A>G, 15:g.27446527C>T, and 15:g.27445825T>C, in the *APOA5* gene was inferred ($R^2$ = 0.99; Figure 2). In the block, the frequencies of the H1 (CTG) and H2 (TCA) haplotypes were 66.7% and 31.7%, respectively. The block was significantly associated with milk yield, fat yield, and protein yield in both lactations (*p* ≤ 0.05; Table 2). H1H1 was the best haplotype for milk yield, and H2H2 was the worst. However, we found no LD for the three SNPs in the *AKT3* gene.

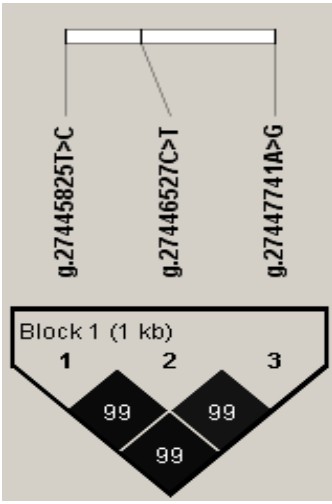

**Figure 2.** Linkage disequilibrium estimated between SNPs in *APOA5* gene. The values in the black boxes are pairwise SNP correlations (R²). The numbers 1, 2 and 3 represent 15:g.27445825T>C, 15:g.27446527C>T and 15:g.27447741A>G respectively.

**Table 2.** Associations of haplotype block in *APOA5* gene with milk production traits in two lactations of Chinese Holstein cows (LSM ± SE).

| Lactation | Haplotype Combination | Milk Yield (kg) | Fat Yield (kg) | Fat Percentage (%) | Protein Yield (kg) | Protein Percentage (%) |
|---|---|---|---|---|---|---|
| 1 | H1H1 (415) | 10,439 Aa ± 64.96 | 345.89 a ± 2.87 | 3.34 ± 0.027 | 308.56 a ± 2.09 | 2.97 ± 0.02 |
| | H1H2 (405) | 10,295 ABb ± 63.30 | 341.90 ab ± 2.80 | 3.33 ± 0.026 | 304.47 b ± 2.04 | 2.97 ± 0.02 |
| | H2H2 (91) | 10,163 Bab ± 94.21 | 337.46 b ± 3.97 | 3.33 ± 0.038 | 301.73 b ± 2.89 | 2.99 ± 0.03 |
| | *p* | 0.0011 | 0.0259 | 0.9935 | 0.0044 | 0.7444 |
| 2 | H1H1 (268) | 10,893 Aa ± 71.27 | 392.86 A ± 3.12 | 3.63 ± 0.029 | 321.60 A ± 2.27 | 2.96 ± 0.02 |
| | H1H2 (279) | 10,649 Bab ± 67.19 | 382.75 B ± 2.97 | 3.61 ± 0.028 | 313.79 B ± 2.16 | 2.96 ± 0.02 |
| | H2H2 (64) | 10,591 ABb ± 110.92 | 385.54 AB ± 4.63 | 3.65 ± 0.045 | 315.26 AB ± 3.38 | 2.99 ± 0.03 |
| | *p* | 0.0005 | 0.0012 | 0.6972 | 0.0005 | 0.5671 |

LSM ± SE is Least Squares Mean ± Standard Error; the number in the bracket represents the number of cows for the corresponding haplotype; genotypes of H1 and H2 are CTG and TCA, respectively; *p* shows the significance for the genetic effects of SNPs; a or b within the same column with different superscripts means $p \leq 0.05$; A or B within the same column with different superscripts means $p \leq 0.01$.

### 3.4. Changes in Transcription Factor Binding Sites Caused by SNPs in 5′ Region

We predicted the changes in TFBSs caused by the three SNPs, 15:g.27447741A>G, 15:g.27446527C>T, and 16:g.33367767T>C, in the 5′ regulatory region of the *APOA5* and *AKT3* genes. For 15:g.27447741A>G in *APOA5*, allele A invented binding sites (BSs) for transcription factor (TF) NK2 homeobox 8 (NKX2.8), and allele G invented BSs for EBF Transcription Factor 1 (EBF1), E74-like ETS transcription factor 5 (ELF5), and HIC ZBTB Transcriptional Repressor 2 (HIC2). Allele C of 15:g.27446527C>T in *APOA5* provided BSs for Twist Family BHLH Transcription Factor 2 (TWIST2) and Rhox Homeobox Family Member 1 (RHOXF1), and when the allele was T there was no BS for any TF. Allele C of 16:g.33367767T>C in *AKT3* created BSs for Nuclear Factor Of Activated T Cells 5 (NFAT5) (Table 3).

**Table 3.** Changes in transcription factor binding sites (TFBSs) caused by the SNPs in 5′ regulatory region of *APOA5* and *AKT3*.

| Gene | SNP Name | Allele | Transcription Factor | Relative Score (≥0.85) | Predicted Core Binding Site Sequence |
|---|---|---|---|---|---|
| *APOA5* | 15:g.27447741A>G | A | NKX2.8 | 0.86 | GCACC<u>T</u>CAG |
| | | G | EBF1 | 0.87 | ACC<u>C</u>CAGGAA |
| | | | ELF5 | 0.86 | C<u>C</u>CAGGAAGAGA |
| | | | HIC2 | 0.88 | GTGCACC<u>C</u>C |
| | 15:g.27446527C>T | C | TWIST2 | 0.86 | CAGA<u>G</u>CTGGG |
| | | | RHOXF1 | 0.88 | CAGA<u>G</u>CTG |
| | | T | - | - | - |
| *AKT3* | 16:g.33367767T>C | T | - | - | - |
| | | C | NFAT5 | 0.87 | ATTTT<u>C</u>TTTT |

Underlined nucleic acids are the SNPs.

*3.5. Changes in mRNA and Protein Structure and Function by Missense Mutation*

We utilized the RNAfold web server to predict the changes in mRNA secondary structure caused by a missense mutation, 15:g.27445825T>C, in the *APOA5* gene and found that when the allele T mutated to C, the MFE changed from −522.90 kcal/mol to −522.80 kcal/mol, indicating that the mRNA secondary structure of this gene is more unstable after mutation. SOPMA analysis revealed that this missense mutation changed the protein secondary structure, with the α-helix changing from 83.70% to 86.96% and random coil from 16.03% to 12.77%, when the allele T mutated into C. By SAAFEC-SEQ prediction, the ΔΔG was reduced to 0.04 kcal/mol after mutation to decrease protein stability. However, this missense mutation was a neutral mutation and did not alter protein function because the predicted PROVEAN score was −0.364. In summary, this missense mutation could reduce the stability of the mRNA secondary structure of the *APOA5* gene and decrease its protein secondary structure and stability.

## 4. Discussion

In GS, SNPs are given different weights based on their importance in the genome relationship matrix, making the prediction of traits more accurate and less biased. For instance, by increasing the weight of SNPs affecting the production performance of Nordic Holstein, Danish Jersey, and Nordic Red cattle, the prediction reliability was increased by up to 3~5% [34]. Sara et al. integrated previously significant SNP information related to the carcass traits of Hanwoo cattle into the GS method, resulting in an improved prediction accuracy of 2~6% [13]. Currently, the six SNPs identified in this study are not present in any of the four gene chips (GeneSeek Genomic Profiler (GGP) Bovine 150K and 100K arrays, illumina Bovine SNP50K BeadChip, illumina BovineHD Genotyping BeadChip). That these SNPs have significant genetic effects on milk production traits suggests that their significant SNPs can be added to the SNP chip, and their weight should be increased during GS to accelerate the selection of cows for milk production traits.

Many phenotypic differences among individuals may be elicited by alterations in gene expression and the underlying transcriptional regulation, and the expression of genes can be regulated by TF binding to TFBSs [35,36]. The SNP located in the TFBS may affect the binding of TF, resulting in differences in gene expression among individuals with different genotypes [37,38]. Here, we found that the SNPs located in the 5′ regulatory regions of the *APOA5* and *AKT3* genes led to changes in gene-binding TFs (Table 3). Studies reported that transcription factors EBF1, ELF5, HIC2, and NFAT5 can promote the expression of target genes to which they bind [39–42], and NKX2.8, TWIST2, and RHOXF1 may inhibit the expression of their target genes [43–46]. The upregulation of *APOA5* can improve the transport of triglycerides from the liver, overall lipid metabolism, and delivery of preformed fatty acids to the mammary gland, thereby promoting the synthesis of milk components [47]. *AKT3* can stimulate β-casein synthesis and mammary epithelial cell proliferation through its involvement in signaling pathways such as mTOR and PI3K/AKT,

thereby promoting milk production traits [48,49]. For instance, we observed that the cows with the GG genotype had significantly higher milk yield, fat yield, and protein yield than those with the AA genotype, suggesting that the GG genotype might activate the expression of the *APOA5* gene by binding the TFs EBF1, ELF5, and HIC2, leading to an increase in milk production traits. Therefore, we inferred that the three SNPs, 15:g.27447741A>G, 15:g.27447741A>G, and 16:g.33367767T>C, alter the TFBS, leading to changes in TF binding, which in turn regulate *APOA5* or *AKT3* gene expression, ultimately affecting milk production traits.

Genetic polymorphisms of alleles can significantly affect the secondary structure of mRNA, and the stability of mRNA largely depends on its secondary structural elements, which will influence the speed and fidelity of its translation into proteins [50]. The alteration in the amino acid sequence leads to changes in the conformation of polypeptide chains, resulting in variations in the protein's secondary structure and influencing protein translation [51,52]. In this study, the milk yield, fat yield, and protein yield of the CC-genotype individuals were significantly higher than those of the TT-genotype individuals, probably because the stability of *APOA5* mRNA and protein decreased when the allele T was mutated to C at 15:g.27445825T>C, suggesting that SNP sites may lead to changes in gene structure and function, and then affect the phenotype.

In this study, we observed that 15:g.27447741A>G, 15:g.27447741A>G, and 15:g.27445825T>C in the *APOA5* gene are in linkage disequilibrium, and haplotype block association analysis revealed that these SNPs have a higher significance in affecting milk yield, fat yield, and protein yield across two lactations compared to single-marker analysis. This may be due to the coordinated effect of these three causal mutations influencing the function or expression of the *APOA5* gene, leading to milk production trait variation. The effects of these SNPs on gene function or expression can be verified by dual luciferase assay, Chromatin Immunoprecipitation (ChIP), Electrophoretic Mobility Shift Assay (EMSA), etc., with which their effects on milk traits can be explored in greater depth.

## 5. Conclusions

This study confirmed the significant genetic effects of three SNPs in the *APOA5* gene and three SNPs in the *AKT3* gene on milk production traits in dairy cattle. Four SNPs were proposed to be the causal mutations affecting milk production traits: three SNPs, 15:g.27447741A>G and 15:g.27446527C>T in *APOA5* and 16:g.33367767T>C in *AKT3*, regulated the expression of genes by alteration of the TFBSs, and one missense mutation in *APOA5*, 15:g.27445825T>C, changed the secondary structure and stability of its mRNA and protein. This project lays a foundation for further functional verification of *APOA5* and *AKT*3, whose valuable SNPs can be used as candidate markers for molecular breeding of dairy cattle.

**Supplementary Materials:** The following supporting information can be downloaded at: https://www.mdpi.com/article/10.3390/agriculture14060869/s1, Table S1: Descriptive statistics of phenotypic values for dairy milk production traits of the first and second lactations; Table S2: Primers and procedures for PCR in SNP identification of *APOA5* and *AKT3* genes; Table S3: Associations of six SNPs in *APOA5* and *AKT3* genes with milk production traits in two lactations of Chinese Holstein cows; Table S4: Additive, dominant, and allele substitution effects of six SNPs on milk production traits of *APOA5* and *AKT3* genes in Chinese Holstein cows.

**Author Contributions:** Conceptualization, D.S.; methodology, Z.G. and L.X.; validation, A.D.; formal analysis, Z.G.; investigation, Z.G., R.T., and W.S.; resources, H.L. and G.Z.; data curation, J.T. and J.Z.; writing—original draft preparation, Z.G.; writing—review and editing, B.H. and D.S.; supervision, X.B.; project administration, D.S.; funding acquisition, D.S. All authors have read and agreed to the published version of the manuscript.

**Funding:** This research was funded by the Science and Technology Program of Inner Mongolia Autonomous Region (2021GG0102; 2021GG0403); the National Key R&D Program of China

(2021YFF1000700); STI 2030—Major Projects (2023ZD04069); and the Program for Changjiang Scholar and Innovation Research Team in University (IRT_15R62).

**Institutional Review Board Statement:** The study was conducted in accordance with the Guide for the Care and Use of Laboratory Animals and approved by the Institutional Animal Care and Use Committee (IACUC) at China Agricultural University (Beijing, China; permit number: DK996, 23 May 2018).

**Data Availability Statement:** The datasets generated and/or analyzed during the current study are available in the article and its Supplementary Material.

**Acknowledgments:** We appreciate Beijing Dairy Cattle Center for providing the semen and blood samples and phenotypic data.

**Conflicts of Interest:** The authors, Wei Sun, Gaoping Zhao and Xiangnan Bao, were employed by the company Inner Mongolia SK Xing Animal Breeding and Breeding Biotechnology Research Institute Co., Ltd., Jixin Zhang was employed by the company Inner Mongolia XuYi Animal Husbandry Co., Ltd. The remaining authors declare that the research was conducted in the absence of any commercial or financial relationships that could be construed as a potential conflict of interest.

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
