# Peer review of "Genetic Association of APOA5 and AKT3 Genes with Milk Production Traits in Chinese Holstein Cows"

_agriculture, doi:10.3390/agriculture14060869_

Round 1
Reviewer 1 Report
Comments and Suggestions for Authors
General comments
Did the authors conduct a power analysis of the tests? Perhaps the number of 944 cows is insufficient to guarantee the reliability of statistical inference. We have as many as 342 subclasses of the HYS effect.
Do not use the "≤" symbol when specifying a specific p-value!
Material and methods
At what age were the cows tested?
Please extract the analysis model and place it on a new line (Line 116).
Subscripts should be included for the specific effects included in the model.
What test was used to assess the significance of differences between the tested genotypes, haplotypes (Tukey, Scheffe, NIR)?
Results
I believe that when describing the results in Tables 2 and 3, it would be worth additionally indicating the best and worst genotypes and haplotypes in terms of milk production traits.
In tables, limit the number of decimal places to 2, except for milk yield.
Discussion
I believe that the results could be discussed in more depth, especially since the authors cite as many as 52 scientific articles.
Conclusions
In my opinion, the "Conclusions" section is basically a summary of the results. Please include in this section a practical conclusion drawn from the research results.
Detailed comments
Lines 25, 27” Should be “P = 0.0146, P = 0.0259)
Line 91: Where can I find Table S1?
Line 103: Where can I find Table S2?
Line 130: A phrase “Jaspar” can be unclear.
Line 163: Should bee “associated with milk yield and protein yields …)
Lines 169-170: What about “protein yields”?
Line 171: Table S3 – Whe can I find this table?
Lines 174, 175: “SE” It means standard error (SE) or standard deviation (SD)
Lies 177, 178: Should be “P ≤ “
Line 188: R2 – unclear.
Reviewer 2 Report
Comments and Suggestions for Authors
Dear Authors,
I have read your article with interest and attention, and in general it seems well presented.
Nonetheless, I have made some notes, in the attached file, which I suggest you consider carefully.
